# Negative/Positive Emotions, Perceived Self-Efficacy and Transition to Motherhood during Pregnancy: A Monitoring Study

**DOI:** 10.3390/ijerph192315818

**Published:** 2022-11-28

**Authors:** Luna Carpinelli, Giulia Savarese

**Affiliations:** Department of Medicine and Surgery, Baronissi Campus, University of Salerno, 84081 Baronissi, Italy

**Keywords:** pregnancy, emotions, self-efficacy, transition to motherhood

## Abstract

Background: Several studies have investigated the topic of emotion regulation and self-perception in women during pregnancy, which turns out to be a critical event for the woman approaching psycho-physical changes. The objectives of the study were the evaluation and monitoring, during pregnancy, of emotional states and levels of self-efficacy and the analysis of the representations of self and the child. Methods: Twenty women (M = 34.60; SD = 4.60) in the 28-week gestation period participated in the research. We performed three administrations (T0-1-2) of an ad hoc questionnaire containing: personal data; Maternal Representations in Pregnancy Interview—IRMAG; Multidimensional Emotion Questionnaire—MEQ; Perceived Self-Efficacy in Complex Situations Scale. Results: Both qualitative and quantitative analyses show that the future mother’s strategies and functional resources focus on perceiving herself as effective in the acquired role, despite the pregnancy itself being a highly stressful critical event. Positive emotions tend to increase, just as the frequency, intensity, persistence and regulation of emotion undergo a linear and constant increase with respect to the first and second administration. Conclusions: Qualitative research has produced significant results with regard to the representations of mothers-to-be as they attempt to cope with states of change during pregnancy with their own personal adaptive resources.

## 1. Introduction

The period between pregnancy and the baby’s first year of life does not always coincide with a peaceful period as the woman preparing to become a mother is involved in a series of changes due both to bodily and physiological modifications and to her own routine and habits [1]. Many studies have examined the variables that can influence the period of pregnancy, trying to investigate the possible effects of stress in this phase of a woman’s life on the development of the foetus and the psycho-physical health of the child after birth and during the first years of life.

A study by Koria et al. [2] investigated how different forms of anxiety and pre-natal stress could increase the risk of emotional or self-regulation difficulties during the first two years of a child’s life. Excessive cortisol production, activated in response to stressful situations (or perceived stressful situations), is associated with respiratory and digestive disorders in children up to the age of three years and also leads to marked behavioural reactivity [3,4]. The foetal period (from the ninth week of gestation to birth) deserves special attention as it is one of the most critical periods for brain development, and in stressful situations, may be more vulnerable to stressful additive factors such as maternal depression [5]. Maternal depression is rather frequent; it affects about 15% of pregnant women (a similar percentage of women in the postnatal period) and can lead to cognitive, behavioural and emotional difficulties in the offspring [6]. Many women may begin to experience a lowered mood and a general perception of inadequacy and decreased levels of self-esteem and self-efficacy from the early stages of pregnancy [7]. Self-efficacy, better known as perceived self-efficacy [8], corresponds to the awareness of being able to master specific activities, situations or aspects of one’s psychological or social functioning. In other words, it is the perception we have of ourselves that we know we are capable of doing, feeling, expressing, being or becoming something.

The scientific literature suggests that self-efficacy functions as a hierarchical organisation of beliefs with different levels of concreteness and complexity of the action to be performed; these beliefs profoundly influence learning and also long-term development [8]. Each belief and its consequences are sensitive to variations in situation, context and task; these beliefs guide and organise each person’s performance and set of actions, which in turn will have positive or negative physical, social and self-esteem consequences.

Questioning one’s parenting skills is very common and, usually, when we find ourselves for the first time embarking on a new task or taking on a new one, it is necessary to reshape our skills. This remodulation, however, involves great effort and commitment and, at times, can put the individual to the test.

Jones and Prinz [9] define parental self-efficacy as “the expectation that caregivers have of their own ability to act successfully as genitors”. Daniel Stern and Nadia Bruschweiler-Stern, in their book dedicated to the changes that accompany the experience of motherhood [10], argue that in order to learn how to be a parent, there are no sufficiently comprehensive guides or suggestions, but it is necessary to try oneself directly in the role, through the tasks and challenges it entails. Therefore, it is understandable that questions about the new parenting tasks are manifold and the fear of failure is high.

Various researchers have tried to give a more specific definition of parental self-efficacy [11]. Some argue that feeling capable coincides with a parent’s general assessment of oneself (“on the whole I am a good parent”). Others, on the other hand, argue that it is more correct to analyse evaluations related to specific tasks, such as “I can feed the child”, “I can get the child to sleep”. Most of the scientific literature agrees with the latter hypothesis, believing that task-specific evaluations structure a parent’s general self-assessment. It has been shown that mothers who perceive themselves as more competent interact more with their children than do mothers who feel less competent. The former show more visual and verbal communication appropriate to their children’s age, physical contact and emotional expressiveness, and seem to understand their children’s requests better and respond appropriately. It would also seem that their children are more likely to explicitly report their needs [12].

### Transition to Motherhood

The changes that occur during pregnancy lead to modifications in the woman’s representational world and consequently, lead to the elaboration of new representations concerning the self as a mother and the future of the child. The co-construction of representations, on the one hand, makes use of perception, which enables the processing of sensory data into ‘schemas’ or ‘maps’; on the other hand, is strongly influenced by the imagination and the role of fantasies and emotions, which influence and connote them by providing particular emotional and fantastical tones [13].

Leff [14] has shown that women during pregnancy develop a maternal style that influences the expectations, fantasies and representations of the pregnant woman and the relationship between mother and child, defining four maternal styles: the ‘regulating’ mother and the ‘facilitating’ mother, the reciprocity style and, finally, the ‘conflictual’ mother. Thus, depending on one’s own experience, mainly positive emotions of joy and hope or lasting and intense negative emotions of anxiety or sadness may emerge. However, even in situations in which the pregnancy is desired and is represented positively in one’s mind, positive and negative emotions, joys and anxieties, hopes and disappointments may alternate. The changes that characterise this sensitive phase of a woman’s life are manifold and the first and immediate one concerns her own body image, which can be difficult for some women to accept.

After childbirth, on the other hand, it is necessary to renounce the state of pregnancy and separate from the inner child, in order to establish an emotional relationship with a real and no longer ideal child. In addition to the physical changes, motherhood also entails consequences on a social and psychological level, since the new mother takes on the responsibilities inherent in the parental role and may sometimes be forced to leave her job, generating financial difficulties in the family; while in other cases, she may fear losing her freedom and identity and may have to reorganise her days according to the child’s needs [15].

The importance noted, therefore, of protecting the mother’s psycho-physical well-being has become an essential condition for ensuring the healthy and harmonious development of the child. These findings, according to scholars, make it necessary to monitor maternal mental health in order to reduce the risk of associated negative outcomes on children’s socio-emotional and behavioural development [7]. On these premises, therefore, the present investigative work aims at: (a) the evaluation of positive and negative emotional states in connection with the perception of self-efficacy; (b) the analysis of the founding themes of self-representation as a mother and of the future image of her child.

## 2. Materials and Methods

### 2.1. Materials

The administration protocol was structured by creating an ad hoc questionnaire according to the variables being researched, as well as an introductory part containing the purpose of the research and informed consent.

The questionnaire was divided into four sections, named and characterised as follows.

Section 1: concerning the subject’s personal data and variables relating to age, marital status, years of relationship, educational qualifications, occupation, number of children.

Section 2: consisting of the Maternal Representations in Pregnancy Interview (IRMAG from the Italian acronym) [16], a semi-structured interview consisting of 41 open-ended questions that can be administered between the 28th and 32nd week of gestation. Specifically, the interview is aimed at detecting self-representation as a mother and the future image of her child.

For the purpose of the present study, 9 open-ended IRMAG questions were selected:Tell me the story of your pregnancy.How come a child at this time in your life?How did you feel when you found out you were pregnant?How was the news received?How did your life change with the pregnancy?Do you think your relationship with your partner has changed?When you realised there was a baby inside you, how did you feel?How do you imagine your child?What kind of mother do you imagine yourself to be?

Section 3: consisting of the Multidimensional Emotion Questionnaire—MEQ [17], a questionnaire investigating two superordinate dimensions of emotional reactivity (positive and negative); three components of positive and negative emotional reactivity, such as frequency, intensity and persistence; ten discrete emotions, classified into: five positive emotions (happiness, excitement, enthusiasm, pride, inspiration) and five negative emotions (sadness, fear, anger, shame, anxiety). The MEQ requests information on personal experience of these emotions and assesses four different components of these emotions. Specifically, for each emotion, it is asked to assess:Frequency with which the emotion is experienced;Intensity of the emotion when it occurs;Duration of the emotion when it occurs;How well can the emotion be regulated when it occurs.

The answer choices for each question are as follows:Frequency with which the emotion is experienced: About once a month, about once a week, about once a day, about 2–3 times a day, more than 3 times a day.Intensity of the emotion when it occurs: very low, low, moderate, high, very high.Duration of emotion when it occurs: less than a minute, 1–10 min, 11–60 min, 1–4 h, more than four hours.How well can the emotion be regulated when it occurs: very easy, easy, moderate, difficult, very difficult.

The MEQ scales are valid and reliable as the Cronbach’s alpha coefficient has good measurement: for the positive regulation (0.76) and negative regulation (0.79) scales were good; also the positive (0.75) and negative (0.79) frequency scales.

Section 4: Consisting of the Perceived Self-Efficacy in Complex Situations Scale [18], which detects subjects’ self-efficacy beliefs in dealing with problematic life experiences. The scale allows for four separate scores for each subject, in relation to each of the factors revealed:Emotional maturity: people’s beliefs about their ability to handle stressful situations; to cope with unexpected events; to have good self-control over difficult events and situations.Finality of action: beliefs that people have about their ability to set concrete and achievable goals, prioritising them and adapting them to their skills, and to pursue the objectives set.Relational fluidity: beliefs that people have about their abilities to interact and deal with others; to give and ask for help, to maintain good relations with others and to manage interpersonal conflicts.Context analysis: people’s beliefs about their ability to ‘read’ the context in which they find themselves operating by grasping the connections between different events and situations; to understand the requests coming from people in the environment; to use language appropriate to different circumstances.

The 4-factor structure of the scale shows good reliability and validity: Cronbach’s alpha coefficients have values ranging from 0.83 to 0.87.

### 2.2. Procedure

Data collection was carried out at 3 different times (T0, T1, T2) in the reference period December 2021–February 2022.

The first meeting (T0) was conducted at the home of 20 pregnant women at 28 weeks’ gestation. During the first meeting, the complete interview proto-collar was administered. Subsequent administrations took place one month (T1) and two months (T2) after the first meeting (T0), using only the MEQ and Self-efficacy scales as monitoring.

For the first administration, the questionnaire was completed in an average time of 30 min. For monitoring, the time taken to write the MEQ and Self-efficacy scales was approximately 13 min.

Prior to administration and during the first meeting, all participating women were provided with informed consent for data processing and the research objective, guaranteeing the subsequent use of the anonymised and aggregated results.

### 2.3. Participants

Twenty women (mean age = 34.60; SD = 4.60) in the 28-week gestation period participated in the research. The variables characterising the sample (see Table 1) showed that 15% are engaged, 25% are cohabiting and 60% are married, and the sample shows an average relationship duration of 6 years (SD = 4.51). With regard to educational qualifications, 5% have a secondary school leaving certificate, 55% have a high school diploma and 40% have a university degree. With regard to employment, 5% are unemployed, 15% are housewives, 15% are freelancers and 65% are employed. With regard to maternity status, it appears that 75% of the women are in their first pregnancy and 25% have already been pregnant with at least one child.

### 2.4. Data Analysis

Both quantitative and qualitative data analyses were carried out.

IBM SPSS v.23 software was used for the quantitative analysis. Frequencies, means and standard deviations were calculated through descriptive statistics and were used to describe the socio-demographic characteristics of the participants and the elements of the questionnaires’ Perceived Self-Efficacy in Complex Situations Scale and MEQ.

ANOVA test was performed to verify significance between T0, T1 and T2 administrations.

For the qualitative analysis, the software T-LAB [19] was used to analyse the most frequent lemmas and their co-occurrences that emerged from the narratives obtained from the IRMAG questionnaire questions. Pre-processing steps included text segmentation, automatic lemmatisation, multiple word detection and selection of key terms. All units of analysis were grouped using a bottom-up or top-down approach.

## 3. Results

### 3.1. Perceived Self-Efficacy in Complex Situations

The averages of the scores obtained in the four dimensions that constitute the Perceived Efficacy in Complex Situations Scale were compared at the different times of administration (see Table 2). From the analysis of the data, it emerges that on the emotional level, the woman’s perceived efficacy is almost completely unchanged over the three administrations, with an average of 18.15 ± 3.60 in T0, 18.20 ± 3.60 in T1 and 18.15 ± 3.60 in T2.

In relation to action, however, there is a tendency of an increase in the woman’s perceived efficacy, mainly at term (T2 = 21.45 ± 4.59), with an average of 21.20 ± 4.73 in T0 and 21.25 ± 4.74 in T1. In the relational context, the woman perceives a self-efficacy that tends to decrease at different times of administration (T0 = 20.70 ± 5.36; T1 = 20.50 ± 5.36; T2= 20.45 ± 5.34). With reference to the context analysis, perceived efficacy tends to decrease at the second administration (T1 = 20.40 ± 3.89) and increases at term (T2 = 20.45 ± 3.92).

### 3.2. Monitoring of Positive and Negative Emotions

The scores obtained from the MEQ scale represent the result of the dimensions characterising the 5 negative emotions, such as sadness, fear, anger, shame, anxiety and 5 positive emotions, such as happiness, excitement, enthusiasm, pride, inspiration. For both poles (positive and negative), averages of the scores for frequency, intensity, persistence, emotional regulation and a general score for negative emotions and positive emotions were identified. A comparison of the mean scores obtained in the emotional dimensions in the three test administrations (see Table 3) shows that the frequency of positive emotions averaged 8.05 ± 4.43 in T0, 9.10 ± 2.93 in T1 and 11.05 ± 2.37 in T2. In this case, the scores for positive emotions are higher than for negative emotions, but the frequency of the latter, although lower, also tends to increase as the pregnancy progresses.

The intensity with which the positive emotion was perceived averaged 8.85 ± 3.34 in T0, 9.80 ± 2.39 in T1 and 11.05 ± 2.39 in T2. As the pregnancy progresses, the emotional intensity increases more and more and the intensity of the negative emotion is stabilised on the same average score, with values of 6.90 ± 3.71 in T0, 6.20 ± 2.82 in T1 and 6.55 ± 2.27 in T2.

Emotion persistence, which corresponds to the permanence of the emotion in the woman, with reference to positivity has an average of 8.20 ± 3.36 in T0, 9.60 ± 2.41 in T1 and 10.70 ± 1.92 in T2. With reference to the negativity of emotion persistence, there is an increase corresponding to an average value of 6.35 ± 3.24 in T0, 5.95 ± 2.60 in T1 and 6.55 ± 2.11 in T2.

In emotion regulation, for positive emotions, women adapt more to the positive state they perceive: in fact, the values are 6.10 ± 2.38 in T0, 7.80 ± 2.37 in T1 and 8.50 ± 2.28 in T2. The same is not true for negative emotions, whose corresponding averages are 5.90 ± 3.68 in T0, 5.70 ± 2.67 in T1 and 5.80 ± 2.28 in T2. In this case, the regulation of negative emotional states is reduced as emotional self-regulation in stressful situations is not efficacy.

The last score obtained is that of general positivity and general emotional negativity, scales derived from the specific emotions analysed (see Table 4). Considering general positivity, the average is 25.10 ± 10.57 in T0, 28.50 ± 7.30 in T1 and 32.80 ± 6.23 in T2. The exponential growth of general positivity corresponds to an increase in general negativity, the average of which has values of 19.25 ± 10.31 in T0, 18.40 ± 8.02 in T1 and in T2 20.35 ± 5.85. Interestingly, as the overall positive emotional effect score increases, so does the negativity score.

### 3.3. Experience of the Self as Mother and Future Child

The participants’ answers and narratives were standardised and classified according to four main themes derived from the content of the IRMAG questions. The single corpus was then subjected to content and word association analysis.

The first main theme is the “perception of pregnancy” (combine the IRMAG responses to questions 1,2,3).

In the main theme, i.e., perception of pregnancy, the word “pregnancy” is the lemma repeated the most times. It is followed by “nausea”, “I”, “years”, “abortion”, “arrived”, “problem” and “first”. The other recurring words are “initially”, “vomit”, “child”, “happy”, “pregnant”, “attempts”, “husband”, “decided”, “child”, “fear”, “immediately”. The smaller the distance between the main point, i.e., “pregnancy”, and the other words, the more these words are associated with the word pregnancy (see Figure 1).

In reference to the main theme, namely, the perception of pregnancy, is represented in the Co-word Analysis, whose corpus is “pregnancy”. At the level of the positive pole, words are found that are associated in a positive way with respect to pregnancy, such as: “pregnant”, “initially”, “abortion”, followed by “attempts”, “happy”, “problem”, “years”, “proceed”, “nausea”, “vomiting” and “before”. At the level of the negative pole, on the other hand, are found words that are associated in a negative way with respect to pregnancy, such as “fear”, “happy”, “son”, “incredulous”, “me”, “child”, “period”, “husband”, “tried”, “immediately”, “decided” and “arrived” who occupies an intermediate position (see Figure 2).

The second main theme is referring to “change” and corresponds to IRMAG answers to questions 4 and 5. The recurring word is again “pregnancy”. The other associations linked to change are “life”, “husband”, “our”, “before”, “family”, “feeling”, “happy”, “son” and “change” (see Figure 3).

In reference to the perception of change, the Co-word Analysis shows that in the upper positive pole, the words “feel”, “change”, “son” and “life” in the middle position are found, while at the lower positive pole are identified the words “family” and “our”.

At the lower negative pole, we find the terms “pregnancy”, “husband” and “before”, while at the upper negative pole “joy” and “happy”. These terms are on the negative pole with respect to the perception of change and assume a lower correlation (see Figure 4).

The third main theme, which is “perception of self and with other persons significant”, combines the answers of the IRMAG questions number 6 and 7.

In this theme, the word “we” is the most repeated lemma. It is followed by the words “movements”, “tune”, “feel” and “grow up”. The other recurring words are “changed”, “pregnancy”, “I”, “relationship”, “husband”, “partner” and “tried”. So the woman speaks of “us”, the word “I” is distant (see Figure 5).

From the Co-word Analysis, it emerges that in the upper positive pole, it is possible to detect the terms “husband”, “tried”, “tune” and “joy”, while in the lower positive pole, the terms “grow”, “pregnancy” and “relationship” are identified. The upper negative pole is characterised by the words “movements”, “feel”, “we”, while in the lower negative pole, we find instead the terms “I”, “partner” and “changed”. What tends to approach the positive pole is the word “we” and “changed” (see Figure 6).

The fourth main theme is “future” and combines IRMAG’s answers to questions 8 and 9. In the main theme “future”, “imagining” is the lemma repeated for the greatest number of times. It is followed by the associations “child”, “eye”, “hair”, “dark”, “beautiful”, “resemble”, “hope”, “father”, “love”, “son” and “mother” (see Figure 7).

The Co-word Analysis (see Figure 8) shows that “imagination” refers to the physical connotations of the child. The upper positive pole consists of terms positively associated with the imagination such as “hair”, “dark” and “beautiful”. In the lower positive pole, the associated terms are “eye”, “imagine”, “child” and “resemble”. The upper negative pole is characterised by the terms “mother”, “affectionate”, “love” and “sweet”, which tend to increase in size to approach the positive pole. The negative pole is lower, instead, and is constituted by the words “son”, “to hope” and “father”.

## 4. Discussion

The results that emerged from our study highlight what is already present in the scientific literature. As regards to perceived self-efficacy, it appears that the future mother’s functional strategies and resources focus on perceiving herself as effective in the acquired role, despite pregnancy itself being a highly stressful critical event [20]. In fact, on an emotional level, the woman’s perceived efficacy is almost completely unchanged over the course of the three administrations, while, in relation to action, there is an increase in the woman’s perceived efficacy, mainly at the end of pregnancy. In the relational context, the woman perceives a self-efficacy that tends to decrease over the different administration times. With reference to the context analysis, perceived efficacy tends to decrease at the second administration, but tends to increase at the end of pregnancy.

With regard to the emotions felt by the woman, it can be seen that general positivity tends to increase as the frequency, intensity, persistence and regulation of the emotion undergo a linear and constant increase with respect to the first and second administration. With regard to general negativity, the frequency and intensity of the emotion increase, while the persistence and regulation of the emotion undergo non-linear variations, as there is an increase at the first administration, a decrease at the second administration and a new increase at the third administration. Just as the overall positive emotional affect score increases, so does the negativity score.

Women are able to adjust to positive emotions [21] whereas they are unable to adjust to negative ones; in fact, there is a decrease in the latter at the second administration, followed by an increase at the third administration.

Qualitative research, on the other hand, has produced significant results with regard to the representations of mothers-to-be, because even before pregnancy sets in, women have an image focused on the self, the problems they have had to face, the attempts they have made to become pregnant and the abortions they have experienced [22]. In fact, the first main theme is defined by the term “pregnancy”, the most frequently associated with which are “nausea”, “self”, “problem” and “abortion”.

Subsequently, these associations tend to evolve as the pregnancy progresses and the perception of it also changes. The second main theme is that of “change” in which the recurring word is still “pregnancy”, thus, as a substantive theme referring to change, the critical event of pregnancy is perceived as such. The other associations are in fact “life”, “our”, “family”, “feeling”. These are associations really related to change, since, in such items and in such responses, the perception changes from woman to mother, as well as the perception of family and life, and the term “our” prevails in contrast to the previously associated term “I”.

The evolution of the associations in relation to pregnancy leads to the third main theme, namely “perception of self and with significant other”, and the concept of “singularity” is replaced by the concept of “plurality”; in fact, in this theme, the word “we” is the most repeated headword. It is followed by the words “movements”, “I feel” and “growing”, which refer to foetal movements and the perception of the foetus as an absent entity. Thus, the woman speaks of “we”, with the word “I” being more distant.

This happens because the woman is focused on what is being constructed, she begins to perceive the movements of the foetus; she thereby achieves a new state of atonement. At the same time, the partner’s conception is as distant as the attempts to achieve pregnancy [23].

Finally, the woman, in the fourth theme “future”, gives space to imagination and turns her thoughts to the future. She investigates the physical features of the child whose presence is increasingly tangible [24,25]. At the same time, she increasingly recognises the transition to the family triad by contemplating the partner as father. The association of the term “love” is more marked than the terms “happiness” and “joy”. As a matter of fact, in the last main theme “future”, the term “imagine” is the most repeated lemma, thus it is represented by the associations “child”, “eye”, “hair”, “father”.

It is important to note that the present study has some limitations: The first of all is the type of sample of “convenience”, so it was not possible to establish some inclusion/exclusion criteria. Secondly, no additional personal and sociodemographic variables of future mothers have been detected, which can affect the Quality of Life and, therefore, emotional states during pregnancy. Finally, there was the possibility of monitoring emotional states and the sense of self-efficacy even after childbirth and within the first year of life of the child, in order to effectively verify the evolution of the mental state of mothers and prevent anxiety-depressive states.

## 5. Conclusions

Pregnancy is a physical, psychological and emotional revolution; it is complex and multifaceted, as are the many and often conflicting emotions experienced by expectant mothers as they prepare to welcome a new little being who will have very specific primary needs and who, from the very first moments of life, will require care, love and a lot of time to devote to the development of the child.

Most expectant mothers experience positive emotions such as joy, enthusiasm and surprise, while others experience completely opposite emotions such as sadness, anger, anxiety, rejection and depression, even before the baby is born.

The joy of sweet expectation is often overwhelmed by negative emotions such as anxiety about something that is unknown, fear of not being able to cope and of not being up to the role of mother. These reactions are not always expressed explicitly by the future mother in order to avoid feeling inadequate with respect to her initial expectations and of being considered unsuitable to be a good mother with respect to the canons and stereotypes of society.

Pregnancy, as well as motherhood, implies both a reworking of one’s role in society, in the couple and in the family, and a mutation of personal identity. Such moments of evolutionary crisis cause an instability of the psyche and it is therefore very important to recognise and welcome all emotions and moods, whether they are positive or negative, to avoid anxiety, fear and sadness turning into more serious psychopathological disorders, such as pre- or post-partum depression.

The expectation of a child disrupts the lifestyle of the mother-to-be and makes it necessary to reorganise the different areas of her life in terms of body, work, relationships, etc., which may make her feel overwhelmed by all these changes to the point of causing a lowering of her mood, an increase in perceived stress levels and the appearance of anxiety symptoms.

The current health and scientific panorama is aiming to organise a good support network that includes the intervention of specialised figures to support mothers-to-be. It is very important that during gestation, care is taken not only to carry out ultrasound examinations, clinical analyses and medical examinations to check that the pregnancy is going well, but also to create a good relationship between mother, father and child from an emotional point of view. From the very first moments after conception, an intense physiological and psychological connection develops between the mother and her child; this implies that a mother who experiences well-being, serenity and joy will transmit these positive feelings to her unborn child; however, if the mother experiences stress, anxiety, fear and/or worry, the baby, even if still in the womb, will be invested by these emotions and these negative feelings will not be beneficial for either of them.

## Figures and Tables

**Figure 1 ijerph-19-15818-f001:**
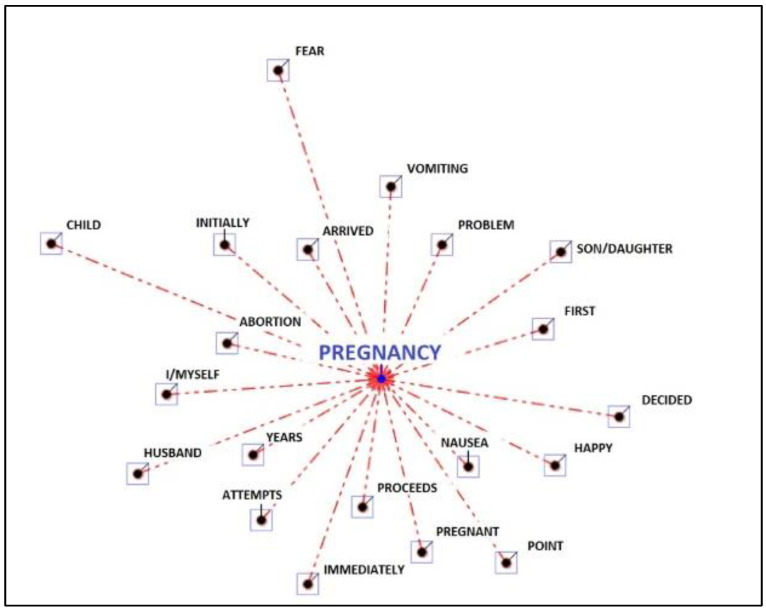
Associations to the word “pregnancy”, referring to the topic “perception pregnancy”.

**Figure 2 ijerph-19-15818-f002:**
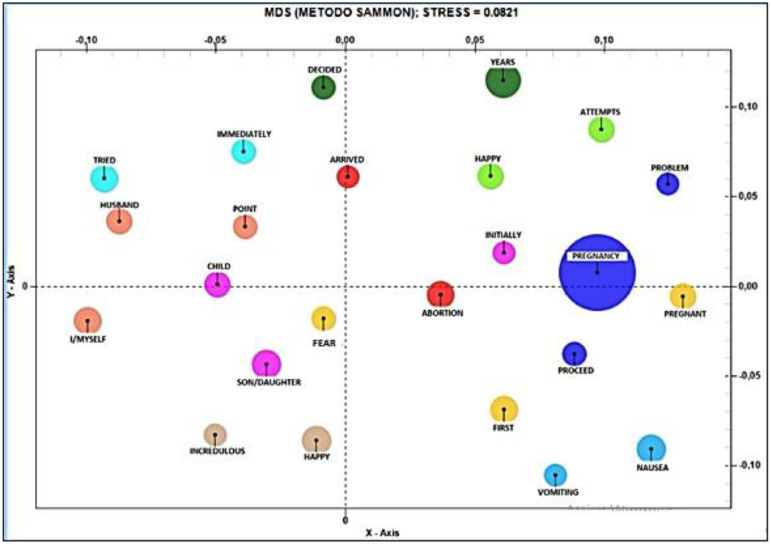
Co-word Analysis referring to the positive/negative positioning of the topic “perception pregnancy”.

**Figure 3 ijerph-19-15818-f003:**
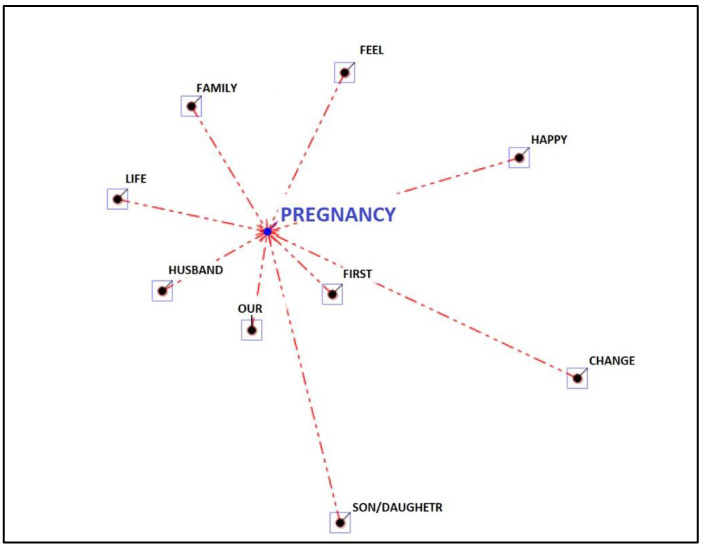
Associations to the topic “change”.

**Figure 4 ijerph-19-15818-f004:**
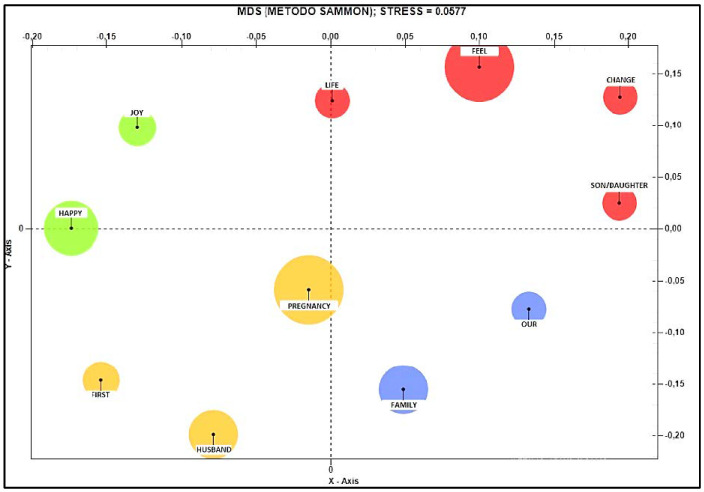
Co-word Analysis referring to the positive/negative positioning of the theme “change”.

**Figure 5 ijerph-19-15818-f005:**
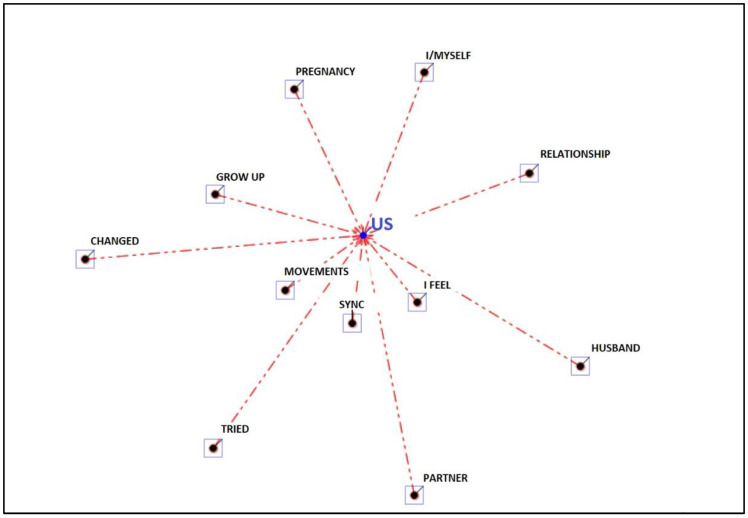
Associations to the theme “perception of self and with other significant”.

**Figure 6 ijerph-19-15818-f006:**
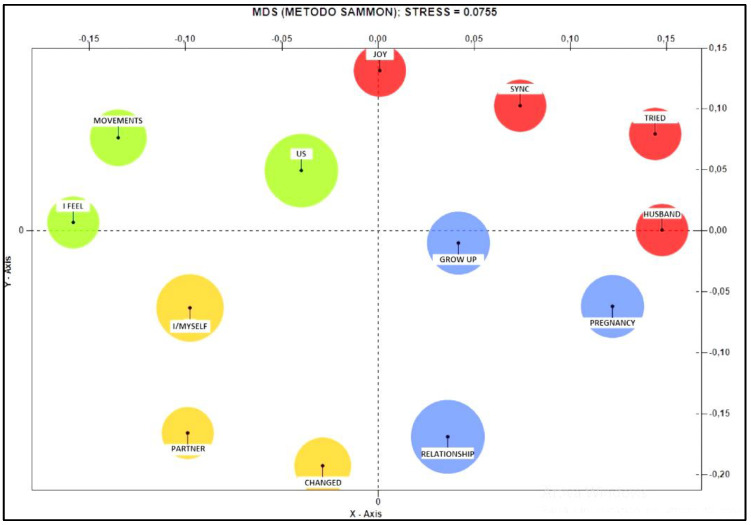
Co-word Analysis referring to the theme “perception of self and with other significant”.

**Figure 7 ijerph-19-15818-f007:**
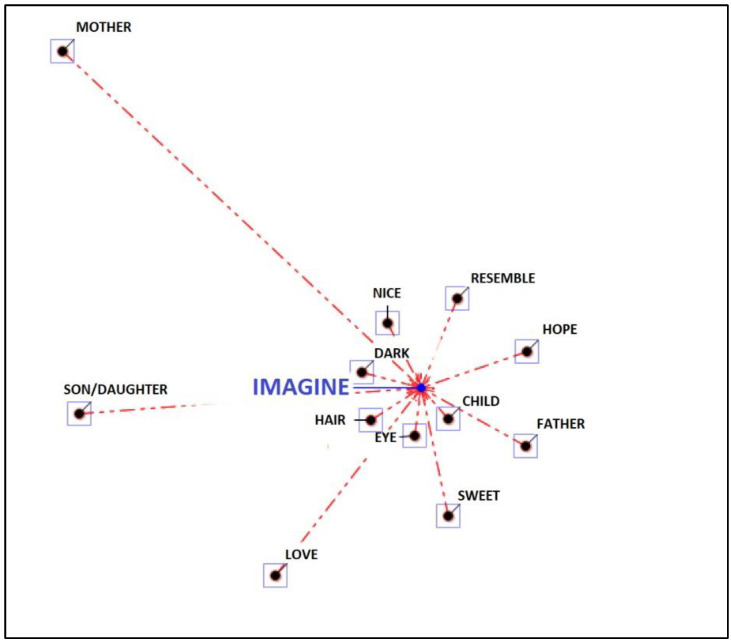
Associations to the theme “future”.

**Figure 8 ijerph-19-15818-f008:**
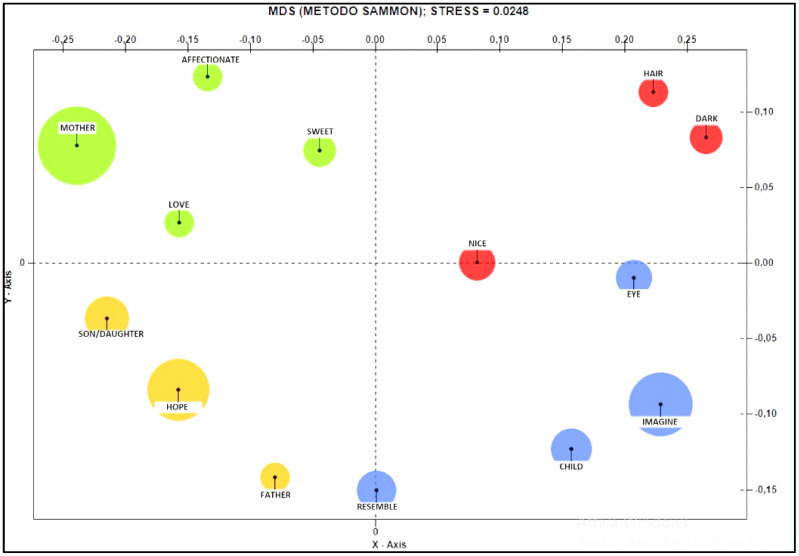
Co-word Analysis referring to the theme “future”.

**Table 1 ijerph-19-15818-t001:** Socio-demographic and work context characteristics of the participants.

Variables	Mean ± Standard Deviation	%
Age	34.60 ± 4.60	-
Marital Status	-	Engaged (15) Married (60) Cohabiting (25)
Years of relationship	6 ± 4.51	-
Educational qualification	-	Secondary school (5) High school (55) University degree (40)
Employment	-	Unemployed (5) Housewife (15) Self-employed (15) Employed (65)
First pregnancy	-	Yes (75) No (25)

**Table 2 ijerph-19-15818-t002:** Means and standard deviations of the Perceived Efficacy in Complex Situations Scale in different times of administration.

Main Category	T0	T1	T2	*p*
Emotion	18.15 ± 3.60	18.20 ± 3.60	18.15 ± 3.60	≤0.05
Action	21.20 ± 4.73	21.25 ± 4.74	21.45 ± 4.59	≤0.05
Relation	20.70 ± 5.36	20.50 ± 5.36	20.45 ± 5.34	≤0.05
Context	20.55 ± 3.83	20.40 ± 3.89	20.45 ± 3.92	≤0.05

**Table 3 ijerph-19-15818-t003:** Means and standard deviations of the MEQ in different times of administration.

**Positive Emotions**	**T0**	**T1**	**T2**
Frequency	8.05 ± 4.43	9.10 ± 2.93	11.05 ± 2.37
Intensity	8.85 ± 3.34	9.80 ± 2.39	11.05 ± 2.39
Persistence	8.20 ± 3.36	9.60 ± 2.41	10.70 ± 1.92
Regulation	6.10 ± 2.38	7.80 ± 2.37	8.50 ± 2.28
**Negative Emotions**	**T0**	**T1**	**T2**
Frequency	6 ± 3.74	6.25 ± 2.95	7.15 ± 2.15
Intensity	6.90 ± 3.71	6.20 ± 2.82	6.55 ± 2.27
Persistence	6.35 ± 3.24	5.95 ± 2.60	6.55 ± 2.11
Regulation	5.90 ± 3.68	5.70 ± 2.67	5.80 ± 2.28

**Table 4 ijerph-19-15818-t004:** Means and standard deviations of the overall positive and negative emotion scores in different times of administration.

**Overall Positive Emotions**	**T0**	**T1**	**T2**	** *p* **
	25.10 ± 10.57	28.50 ± 7.30	32.80 ± 6.23	<0.001
**Overall Negative Emotions**	**T0**	**T1**	**T2**	** *p* **
	19.25 ± 10.31	18.40 ± 8.02	20.35 ± 5.85	<0.001

## Data Availability

Written informed consent was obtained from the subjects in order to publish this paper. The archived data is not public and can be requested by writing to the corresponding author.

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
