# Peer review of "Negative/Positive Emotions, Perceived Self-Efficacy and Transition to Motherhood during Pregnancy: A Monitoring Study"

_ijerph, 2022, doi:10.3390/ijerph192315818_

Round 1

Reviewer 1 Report

Dear author, 

Thank you for working on interesting topic. Here are some points to be considered to improve the quality of your manuscript:

1)      A couple of spaces are missing in the Abstract

2)      Methods: How you justify the frequency used in questionnaire? (lines 162-163):

“Duration of emotion when it occurs: less than a minute, 1-10 minutes, 11-60 162 minutes, 1-4 h, more than four hours.”

3)      It is a typo error or why are using capital letters “IN THE” (line 185):

Data collection was carried out at 3 different times (T0, T1, T2) IN THE reference 185 period December 2021 - February 2022.

4)      As per increase and decrease, you refer to crue data? Or any analysis of statistically significant differences? It would potentiate your results if you perform it (Lines 229-234):

“In relation to action, however, there is an increase in the woman's perceived efficacy, 229 mainly at term (T2= 21.45±4.59), with an average of 21.20±4.73 in T0 and 21.25±4.74 in T1. 230 In the relational context, the woman perceives a self-efficacy that tends to decrease at different times of administration (T0= 20.70±5.36; T1= 20.50±5.36; T2= 20.45±5.34). With reference to the context analysis, perceived efficacy tends to decrease at the second administration (T1= 20.40±3.89) and increases at term (T2= 20.45±3.92).”

Or at least use “tendencies” everywhere then.

5)      Table 2 (and other places): you use means – are the data normally distributed? In case if not, it is also good to add a column of “mode” to see the group tendency for each variable.

6)      Graphs 1, 2, 4, 6 & 8 – Please, increase letter size since it is too small to read the words used there.

7)       Study Limitations should be included in the Discussion section.

Author Response

Comments and Suggestions for Authors

Authors: The Authors thank the Reviewer for the valuable suggestions.

Dear author,

Thank you for working on interesting topic. Here are some points to be considered to improve the quality of your manuscript:

1)      A couple of spaces are missing in the Abstract

Authors: done

2)      Methods: How you justify the frequency used in questionnaire? (lines 162-163):

“Duration of emotion when it occurs: less than a minute, 1-10 minutes, 11-60 162 minutes, 1-4 h, more than four hours.”

Authors: This specification is part of the characteristics of the test we used. The MEQ is a validated and standardised instrument. We have inserted the statistical details of Cronbach’s alpha coefficient of the MEQ.

3)      It is a typo error or why are using capital letters “IN THE” (line 185):

Data collection was carried out at 3 different times (T0, T1, T2) IN THE reference 185 period December 2021 - February 2022.

Authors: correct “in the”.

4)      As per increase and decrease, you refer to crue data? Or any analysis of statistically significant differences? It would potentiate your results if you perform it (Lines 229-234):

“In relation to action, however, there is an increase in the woman's perceived efficacy, 229 mainly at term (T2= 21.45±4.59), with an average of 21.20±4.73 in T0 and 21.25±4.74 in T1. 230 In the relational context, the woman perceives a self-efficacy that tends to decrease at different times of administration (T0= 20.70±5.36; T1= 20.50±5.36; T2= 20.45±5.34). With reference to the context analysis, perceived efficacy tends to decrease at the second administration (T1= 20.40±3.89) and increases at term (T2= 20.45±3.92).”

Or at least use “tendencies” everywhere then.

Authors: we changed with terms “tendencies”.

5)      Table 2 (and other places): you use means – are the data normally distributed? In case if not, it is also good to add a column for “mode” to see the group tendency for each variable.

Authors: Yes, there is a normal distribution. We have specified in the “Results” paragraph.

6)      Graphs 1, 2, 4, 6 & 8 – Please, increase letter size since it is too small to read the words used there.

Authors: We reshaped the background/figure brightness contrasts and increased the font size.

7)       Study Limitations should be included in the Discussion section.

Authors: We have inserted the limitations in the paragraph "Discussion".

Reviewer 2 Report

Abstract

·         Lead in statement didn’t link the aims of the study

·         Research design should be mentioned as it didn’t provide an overview of how the study was being conducted.

Introduction

·         Line 33, clarify which is issue ‘this’ is referring to.

·         Avoid double negatives like line 44 “not so infrequent”.

·         “Many women may begin to experience a lowered mood and a general perception of inadequacy and decreased levels of self-esteem and self-efficacy from an early age.” Is the decreased levels referring to the mothers or children? This is not clear.

·         The discussion around self-efficacy is not quite clear. Do you mean domain specific vs general self-efficacy? If so, stick to these terms as they are the usual terminology used.

·         “Some are immediate, but others require effort and can put a strain on the individual.” Do ‘some’ and ‘others’ refer to skills the need to be acquired or reviewed? If so, do you mean that some of these skills are easily attained with minimal effort, whereas others are more complex and take time to learn or acquire? This is not explained clearly.

·         Some of the discussion around parental self-efficacy seem to focus on parents of older children rather than infants. This doesn’t seem to align with the target population the study is focusing on.

·         The argument in the introduction is hard to follow. The paragraphs, points and ideas need to be synthesised and expressed better.

Methods

·         Citation needed for scales (MEQ & Perceived self-efficacy)

·         Have the scales been used previously and report the psychometric properties of the questionnaire?

·         Why wasn’t a self-efficacy measure more specific to parenting chosen instead, given that you had argued that task-specific measures are better at providing general self-assessment?

·         Given that some of the participants are not first time mothers, wouldn’t that have an impact on their perceived self-efficacy as a parent?

·         Why wasn’t any statistical test done for the repeated measures?

·         Results

·         Without statistical testing, it would be inappropriate just to comment on means. However, even if statistical tests were run, the results would be very much underpowered.

·         Graph 2 & 4 were hard to understand. It would be better to label the graphs with labels such as “Positive” and “negative” so that it is easier to orientate to.

Discussion

·         The points discussed relating to the quantitative data were overstated given that no statistical analysis was being conducted.

·         The points raised do not add insight into the current literature, and seem to be descriptive rather than informative. The implications of the findings need to be clear.

·         Limitations is missing.

Author Response

Comments and Suggestions for Authors

Authors: Thank you for your valuable suggestions and the time you took to improve our paper.

Abstract

  • Lead in statement didn’t link the aims of the study
  • Research design should be mentioned as it didn’t provide an overview of how the study was being conducted.

Authors: We redefined the abstract based on the suggestions.

Introduction

  • Line 33, clarify which is issue ‘this’ is referring to.

Authors: We modified as follow: “Many studies have examined the variables that can influence the period of pregnancy,”

  • Avoid double negatives like line 44 “not so infrequent”.

Authors: We modified in “Maternal depression is rather frequent”

  • “Many women may begin to experience a lowered mood and a general perception of inadequacy and decreased levels of self-esteem and self-efficacy from an early age.” Is the decreased levels referring to the mothers or children? This is not clear.

Authors: Abbiamo modificato “Early stage” into “from the early stages of pregnancy”

  • The discussion around self-efficacy is not quite clear. Do you mean domain specific vs general self-efficacy? If so, stick to these terms as they are the usual terminology used.

Authors: We have reshaped the terminology as suggested.

  • “Some are immediate, but others require effort and can put a strain on the individual.” Do ‘some’ and ‘others’ refer to skills the need to be acquired or reviewed? If so, do you mean that some of these skills are easily attained with minimal effort, whereas others are more complex and take time to learn or acquire? This is not explained clearly.

Authors: We have reshaped the sentence as follow: “Questioning one’s parenting skills is very common and, usually, when we find ourselves for the first time embarking on a new task or taking on a new one it is necessary to reshape our skills. This remodulation, however, involves great effort and commitment and, at times, can put the individual to the test.”

  • Some of the discussion around parental self-efficacy seem to focus on parents of older children rather than infants. This doesn’t seem to align with the target population the study is focusing on.
  • The argument in the introduction is hard to follow. The paragraphs, points and ideas need to be synthesised and expressed better.

Methods

Authors: We have reviewed both "Introduction" and "Discussion" and all this is reportedly within the period of pregnancy.

  • Citation needed for scales (MEQ & Perceived self-efficacy)

Authors: done

  • Have the scales been used previously and report the psychometric properties of the questionnaire?

Authors: yes, We have inserted validity and reliability measures .

  • Why wasn’t a self-efficacy measure more specific to parenting chosen instead, given that you had argued that task-specific measures are better at providing general self-assessment?

Authors: Thank you for the question, important moment of reflection. The choice of the scale used is because in our background we specified the period of pregnancy as a critical event for the woman/future mother. For this reason and as already present in the literature, we used a scale of self-efficacy that investigated at that specific moment the perception of self of the future mother. In addition, parental-specific self-efficacy scales measure the construct when the couple has already had their child.

  • Given that some of the participants are not first time mothers, wouldn’t that have an impact on their perceived self-efficacy as a parent?

Authors: We put it as a limitation.

  • Why wasn’t any statistical test done for the repeated measures?

Authors: We carried out ANOVA to repeated measurements.

  • Results
  • Without statistical testing, it would be inappropriate just to comment on means. However, even if statistical tests were run, the results would be very much underpowered.

Authors: We put it as a limitation.

  • Graph 2 & 4 were hard to understand. It would be better to label the graphs with labels such as “Positive” and “negative” so that it is easier to orientate to.

Authors: We specified positive/negative positioning in the label.

Discussion

  • The points discussed relating to the quantitative data were overstated given that no statistical analysis was being conducted.
  • The points raised do not add insight into the current literature, and seem to be descriptive rather than informative. The implications of the findings need to be clear.

Authors: We have reshaped these suggestions in the "Discussion" section to better highlight the impact of our results.

  • Limitations is missing.

Authors: We have inserted the limitations in the paragraph "Discussion".

Round 2

Reviewer 2 Report

Thank you for your responses to my comments.

Please include the p-values for the Perceived Self-Efficacy in Complex Situations Scale in the results section Table 2.

Author Response

We have  included the p-values for the Perceived Self-Efficacy in Complex Situations Scale in the results section Table 2.
